# Region-wise Motion Controller for Image-to-Video Generation

## Abstract

Animating images with interactive motion control has garnered popularity for image-to-video (I2V) generation. Modern approaches typically regard the condition of Gaussian filtered point-wise trajectory as sole motion control signal. Nevertheless, such flow approximation of trajectory via Gaussian kernel severely limits the controllable capacity of fine-grained movement, and commonly fails to disentangle object and camera moving. To alleviate these, we present ReMoCo, a new recipe of region-wise motion controller that novelly leverages precise region-wise trajectory and motion mask to regulate fine-grained motion synthesis and identify exact target motion category (i.e., object or camera moving), respectively. Technically, ReMoCo first estimates the flow maps on each training video via a tracking model, and then samples the region-wise trajectories from multiple local regions to simulate inference scenario. Instead of approximating flow distribution via Gaussian filtering, our region-wise trajectory preserves original flow information at local area and thus manages to characterize fine-grained movement. A motion mask is simultaneously derived from the predicted flow maps to present holistic motion dynamics. To pursue natural and controllable motion generation, ReMoCo further strengthens video denoising with additional conditions of region-wise trajectory and motion mask in a feature modulation manner. More remarkably, we meticulously construct a benchmark called *ReMoCo-Bench*, which consists of $1.1K$ real-world user-annotated image-trajectory pairs, for the evaluation of both fine-grained and object-level motion synthesis in I2V generation. Extensive experiments conducted on WebVid-10M and ReMoCo-Bench demonstrate the effectiveness of our ReMoCo for precise motion control.

## 1 Introduction

In recent years, diffusion models (Ho et al., 2022a; Blattmann et al., 2023b; Singer et al., 2023; Ge et al., 2023; Brooks et al., 2024) have shown significant progress in revolutionizing text-to-video (T2V) generation. Although promising visual appearance can be attained by these advances, the controllable motion generation is still a grand challenge in video diffusion paradigm. There are several attempts (Esser et al., 2023; Wang et al., 2023; Chai et al., 2023) to enhance controllable capacity of video synthesis with additional guidance (e.g., depth, edge or optical flow). Nevertheless, it might be impractical for users to conveniently provide such signals as input conditions. Hence, the focus of this paper is to capitalize on the user-friendly conditions (i.e., sparse trajectory and region mask) for enabling interactively controllable image-to-video (I2V) generation: given the reference image as the first frame, the motion in the synthesized video should be natural and well-aligned with the provided trajectory.

Pioneering practices (Yin et al., 2023; Wu et al., 2024) of controllable I2V generation usually guide video denoising process with the single condition of Gaussian filtered trajectory. In the training stage, the input trajectories are first sparsely sampled from the optical flow maps and then processed by Gaussian filter. The flow approximation brought by Gaussian filtering inevitably results in the inaccuracy of fine-grained motion details and limits the model capability for precise motion control. Therefore, the generated fine-grained movement (e.g., the turning-head of first case in Figure 1) is unnatural. Another issue is that the single condition of trajectory commonly fails to precisely identify the target motion category (i.e., camera or object moving). For instance, as depicted in Figure 1, the trajectory on the planet could be explained as two moving situations, i.e., the camera

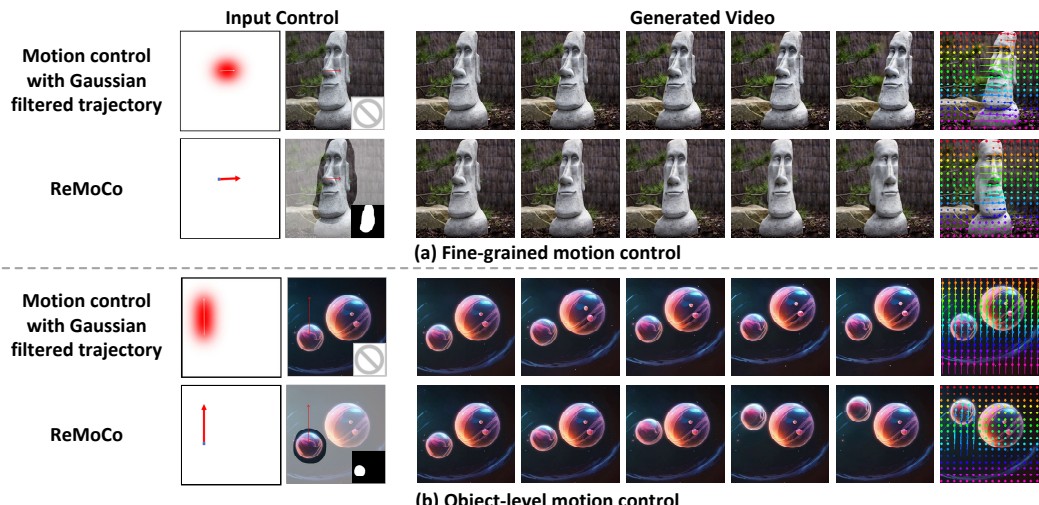

Figure 1: An illustration of (a) fine-grained and (b) object-level motion control by using typical Gaussian filtered trajectory and our region-wise motion controller (ReMoCo). The trajectories of generated videos are visualized in the last frame.

being pulled downwards with relative to static two planets (camera movement) or planet rising corresponding to static background (object movement). Solely relying on the trajectory might lead to the motion misinterpretation and thus hinder exactly controllable I2V generation. To address the above two issues, we shape a new paradigm of motion controller that capitalizes on region-level trajectory and motion region mask to enhance video denoising for controllable motion synthesis. Specifically, we spatially sample multiple local regions in the video optical flow maps and directly employ the trajectories in the sparse regions as input trajectory condition. In this way, no flow approximation is included in such region-wise trajectory, which manages to adequately reflect the local fine-grained motion details. Meanwhile, a region mask is estimated on the video optical flow maps which aims to globally emphasize the motion area, thereby specifying the target motion category and alleviating misinterpretation. To further regulate the motion synthesis in I2V generation, we predict the affine parameters on the collaboration of trajectory and motion mask to modulate the video latent codes during denoising. As shown in Figure 1, our unique region-wise trajectory design and the employment of motion mask complementarily achieves the better fine-grained (e.g., turning-head) and object-level (e.g., planet-moving) motion generation.

By materializing the idea of facilitating controllable I2V generation with the proposed conditions, we present a novel framework, namely ReMoCo, to execute Region-wise Motion Control. Specifically, given the input video, ReMoCo first estimates the sequence of visibility masks and optical flow maps by using an off-the-shelf optical tracking model. Next, the global visibility mask is obtained through computing the intersection of all visibility masks, and further multiplied with the flow map of each frame. Then, ReMoCo splits the masked flow maps into multiple local regions (e.g., the region with the size of $8 \times 8$) and employs the trajectories on such sparsely-sampled regions as region-wise trajectory. Meanwhile, ReMoCo attains the motion mask on the flow maps via thresholding mechanism for representing holistic motion. Given the region-wise trajectory and corresponding motion mask, the multi-scale features are learnt by a motion encoder, and further employed to predict scale and bias for video latent feature modulation. Moreover, ReMoCo fine-tunes all attention modules in 3D-UNet via utilizing the Low-Rank Adaptation (LoRA) technique to pursue better motion-trajectory alignment.

In summary, we have made the following contributions:

- We introduce a new design of region-wise trajectory and motion mask as the complementary control signals in I2V diffusion models for the interactive motion control.
- A novel approach, namely Region-wise Motion Controller (ReMoCo), seamlessly integrates the proposed region-wise trajectory and motion mask into 3D-UNet to guide video denoising for natural and precise motion synthesis in I2V generation.

- We present *ReMoCo-Bench*, to our best knowledge, which is one of the first benchmarks with real-world user-annotated image-trajectory pairs for controllable I2V generation. Extensive experiments on WebVid-10M and ReMoCo-Bench verify the superiority of ReMoCo in terms of both video quality and motion-trajectory alignment.

## 2 RELATED WORK

**Image-to-Video Diffusion Models.** The remarkable progress achieved by text-to-video generation (Ho et al., 2022b;a; Blattmann et al., 2023b; Khachatryan et al., 2023; Luo et al., 2023; Singer et al., 2023; Ge et al., 2023; Gupta et al., 2023; Guo et al., 2024; Brooks et al., 2024) encourages the development of image-to-video (I2V) diffusion models. These advances (Girdhar et al., 2024; Blattmann et al., 2023a; Xing et al., 2024; Shi et al., 2024a; Zeng et al., 2024) treat static image as the input condition for temporal coherent video synthesis. VideoComposer (Wang et al., 2023) is one of the earlier works that integrates image condition into 3D-UNet through concatenating the clean image latent with the noisy video latents. Based on this recipe, DynamiCrafter (Xing et al., 2024) and SVD (Blattmann et al., 2023a) additionally inject the CLIP (Radford et al., 2021) feature of reference image into video denoising to enhance the information guidance. To achieve high-resolution I2V generation, I2VGen-XL (Zhang et al., 2023b) introduces a cascading diffusion model to first animate image in the low resolution and further magnifies it via video refinement. Besides, there are several explorations (Chen et al., 2023b; Zeng et al., 2024) that simultaneously utilize two images (i.e., the first and last frames) as more powerful references to elevate I2V generation. In this work, we choose the pre-trained I2V diffusion model SVD (Blattmann et al., 2023a) as our base architecture for motion control.

**Controllable Video Diffusion Models.** Despite high-quality video synthesis via I2V diffusion models, the controllable motion generation still remains an under-explored problem. The early controllable video diffusion techniques (Wang et al., 2023; Esser et al., 2023; Chen et al., 2023a; Zhang et al., 2024) typically leverage the condition of depth, edge or optical flow, for particular motion generation. Nevertheless, it is usually impractical for users to conveniently obtain such kinds of signals. To address this issue, the studies exploring bounding box (Jain et al., 2024; Wang et al., 2024a) or trajectory (Yin et al., 2023; Wu et al., 2024; Niu et al., 2024; Mou et al., 2024; Wang et al., 2024b) as additional condition for motion control start to emerge. One representative of using bounding box as control is PEEKABOO (Jain et al., 2024) which designs the training-free spatial-temporal masked attention for visual-textual alignment in the box. In the direction of utilizing trajectory condition, pioneering advance DragNUWA (Yin et al., 2023) exploits Gaussian filtered trajectory to regulate motion synthesis via multi-scale feature fusion. Wu et al. (2024) further incorporate the entity features of reference image into diffusion to facilitate object-level motion control. Recently, MOFA-Video (Niu et al., 2024) devises a two-stage motion control framework that first densifies input trajectories via conditional motion propagation (CMP), and further regulates video denoising with the estimated dense trajectories. Nevertheless, most of the existing works employ the Gaussian filtered trajectory as the single condition. The Gaussian filtering will lead to flow approximation in local area, which constrains the capacity for fine-grained motion modeling. Solely capitalizing on trajectory could also fail to disentangle object and camera moving in I2V motion synthesis.

In short, our work mainly focuses on a new recipe of motion condition, i.e., the region-wise trajectory and motion mask, and the exploitation of these conditions for exact controllable I2V generation. The proposal of ReMoCo contributes by studying not only how to express the motion trajectory accurately, but also how to benefit natural and precise motion generation with the synergy of the region-wise trajectory and motion mask.

## 3 OUR APPROACH

In this section, we introduce our Region-wise Motion Controller (ReMoCo) for controllable I2V generation. Figure 2 illustrates an overview of our ReMoCo. Given a video clip at training, the newly-minted region-wise trajectory and motion mask are first extracted as the control signals. Next, multi-scale features are learnt on the concatenation of the trajectory and mask via a motion encoder. These features are further injected into the 3D-UNet of SVD (Blattmann et al., 2023a) to regulate video denoising. In each feature scale of the 3D-UNet, a scale and bias are predicted through

Figure 2: An overview of our Region-wise Motion Controller (ReMoCo) for controllable image-to-video generation. During training, ReMoCo first extracts the proposed region-wise trajectory and motion mask on the input video as the control signals. The multi-scale features are then learnt on these signals by a motion encoder, and further injected into the 3D-UNet of SVD in a feature modulation manner. Meanwhile, LoRA layers are integrated into all attention modules in the transformer blocks to improve the optimization of motion-trajectory alignment. In the inference stage, the region-wise trajectory and motion mask are first derived from the user provided trajectory and brushed region, and then exploited as the guidance to calibrate I2V video generation.

convolutional layers to modulate the feature of video latent codes. Besides, all attention modules are fine-tuned by LoRA (Hu et al., 2022) to attain better alignment between the synthesized motion and input trajectory.

## 3.1 PRELIMINARIES: STABLE VIDEO DIFFUSION

To leverage comprehensive motion prior embedded in the pre-trained diffusion models for video generation, we exploit the advanced I2V generation model, i.e., Stable Video Diffusion (SVD) (Blattmann et al., 2023a) as the base architecture of our ReMoCo. To better understand our proposal, we first revisit the training procedure of SVD. Formally, given an input video clip $\mathbf{x}_0 = \{x_0^i\}_{i=1}^L$ with $L$ frames, the clean video latent codes $\mathbf{z}_0 = \{z_0^i\}_{i=1}^L$ are first extracted via a variational auto-encoder (VAE). Then, the Gaussian noise $\mathbf{n}$ is added to $\mathbf{z}_0$ through forward diffusion procedure as:

$$\mathbf{z} = \mathbf{z}_0 + \mathbf{n}, \quad (\sigma, \mathbf{n}) \sim p(\sigma, \mathbf{n}), \tag{1}$$

where $\mathbf{z}$ is the noised video latent codes and $p(\sigma, \mathbf{n}) = p(\sigma)\mathcal{N}(\mathbf{0}, \sigma^2\mathbf{I})$. $\sigma$ represents the noise level and $p(\sigma)$ is the pre-determined distribution over $\sigma$. Following the training protocol of EDM (Karras et al., 2022), SVD leverages the 3D-UNet $F_{\boldsymbol{\theta}}$ (with parameters $\boldsymbol{\theta}$) to predict the clean video latent codes $\hat{\mathbf{z}}_0$ with the condition of input noised latents $\mathbf{z}$, noise level $\sigma$ and the reference image $\mathbf{c}_I$:

$$\hat{\mathbf{z}}_0 = c_{\text{skip}}(\sigma)\mathbf{z} + c_{\text{out}}(\sigma)F_{\boldsymbol{\theta}}(c_{\text{in}}(\sigma)\mathbf{z}, \mathbf{c}_I; c_{\text{noise}}(\sigma)), \tag{2}$$

where $c_{\text{skip}}(\sigma)$, $c_{\text{out}}(\sigma)$, $c_{\text{in}}(\sigma)$ and $c_{\text{noise}}(\sigma)$ are pre-defined hyper-parameters determined by noise level $\sigma$. In SVD, the information of reference frame is injected into 3D-UNet along two pathways: a) the channel-wise concatenation of noised video latent codes and first frame latent code; b) the cross-attention between video latent feature and image CLIP (Radford et al., 2021) embedding of first frame. The loss function is formulated via denoising score matching (DSM) as:

$$\mathcal{L} = \mathbb{E}_{(\mathbf{z}_0, \mathbf{c}_I) \sim p_{\text{data}}(\mathbf{z}_0, \mathbf{c}_I), (\sigma, \mathbf{n}) \sim p(\sigma, \mathbf{n})} \left[ \lambda_\sigma \|\hat{\mathbf{z}}_0 - \mathbf{z}_0\|_2^2 \right], \tag{3}$$

where $\lambda_\sigma$ is a weighting function. In the scenario of our work, besides the condition of reference first frame, we additionally exploit a new kind of region-wise trajectory and motion mask as the control signals to refine video denoising for motion control.

## 3.2 MOTION CONDITION GENERATION

Most existing controllable I2V approaches calibrate the video denoising with the sole guidance of Gaussian filtered point-wise trajectory. Nevertheless, the flow approximation brought by Gaussian filtering may result in inaccuracy of fine-grained motion details. Therefore, the ability of precise motion control could be limited. Besides, solely relying on the trajectory for motion control might not exactly express target motion category (i.e., camera or object moving), leading to motion misinterpretation in video generation. To alleviate these issues, we propose to directly sample trajectories from optical flow maps in multiple local regions as the region-wise trajectory. Such trajectory preserves the original flow information in local regions, and thus manage to characterize fine-grained movement. In the meanwhile, a motion mask is further derived from the flow maps to explicitly identify target motion category of the generated videos.

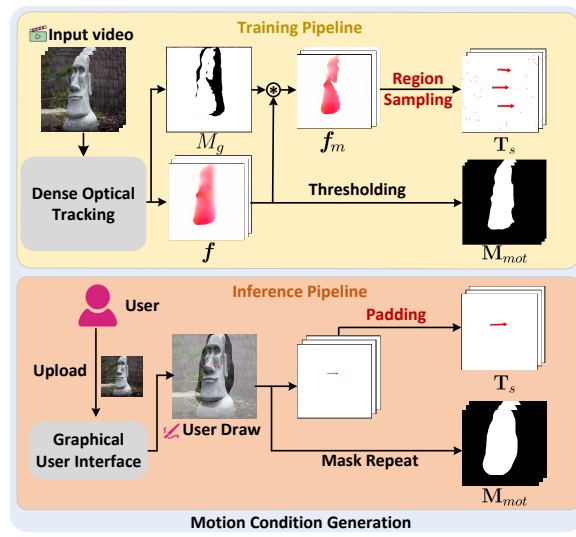

Figure 3: Motion condition generation in the training and inference stages of our ReMoCo.

**Region-wise Trajectory.** As depicted in Figure 3, given the input video clip $\mathbf{x}_0 = \{x_0^i\}_{i=1}^L$ with the size of $L \times H \times W \times 3$, we first employ a dense optical tracking model, i.e., DOT (Moing et al., 2024) to estimate optical flow maps $\boldsymbol{f} = \{f^i\}_{i=1}^L$ and the sequence of visibility masks $\mathbf{M} = \{M^i\}_{i=1}^L$:

$$f^i, M^i = \text{DOT}(\mathbf{x}_0^1, \mathbf{x}_0^i), \quad i = 1, 2, ..., L, \tag{4}$$

where $f^i \in \mathbb{R}^{H \times W \times 2}$ and $M^i \in \{0, 1\}^{H \times W}$ is the optical flow map and the visibility mask between the first and the $i$-th frame, respectively. Then, we calculate the intersection on $\mathbf{M}$ to attain a global visibility mask $M_g \in \{0, 1\}^{H \times W}$ that indicates the locations having visible optical flow along temporal dimension as:

$$M_g = \prod_{i=1}^L M^i. \tag{5}$$

Next, the masked flow maps $\boldsymbol{f}_m = \{f_m^i\}_{i=1}^L$ are computed by frame-wisely multiplying the flow maps $\boldsymbol{f}$ with the global visibility mask $M_g$ as follows:

$$\boldsymbol{f}_m = \{f^i \cdot M_g\}_{i=1}^L. \tag{6}$$

We split the masked flow maps $\boldsymbol{f}_m$ into multiple local regions and the spatial size of each region is $k \times k$. The region-wise trajectories $\mathbf{T}_s \in \mathbb{R}^{L \times H \times W \times 2}$ are finally sampled from the region-split $\boldsymbol{f}_m$ with a region selection mask $M_{sel} \in \{0, 1\}^{\frac{H}{k} \times \frac{W}{k}}$:

$$\mathbf{T}_s = \{f_m^i \cdot Pad(M_{sel})\}_{i=1}^L, \tag{7}$$

where $M_{sel}$ is uniformly sampled from $\{0, 1\}$ with the mask ratio $r_m$, and $Pad(\cdot)$ denotes the padding function which fills the mask value into the $k \times k$ region around each position. Instead of exploiting a constant mask ratio for trajectory selection, we randomly choose $r_m$ in a range of $[r_{min}, 1.0]$ to simulate different real-world motion masking scenarios, which benefits the robust network optimization. In this way, there is no flow approximation of the trajectories in each local region, enhancing the control ability of fine-grained motion in I2V models.

**Motion Mask.** In addition to the region-wise trajectory for video denoising regulation, the motion mask aims to specify the motion category and benefit the global motion correlation. Given the flow maps $\boldsymbol{f} = \{f^i\}_{i=1}^L$ estimated by DOT, we first calculate the average flow magnitude $f_{avg} \in \mathbb{R}^{H \times W}$ along temporal dimension as: $f_{avg} = \frac{1}{L} \cdot \sum_{i=1}^L \| f^i \|_2$. Then, we construct the motion mask $M_{mot} \in \{0, 1\}^{H \times W}$ from zero matrix, and set the value of the position where $f_{avg}$ is greater than 1 as True. $M_{mot}$ is finally repeated $L$ times as the motion mask sequence $\mathbf{M}_{mot} \in \{0, 1\}^{L \times H \times W \times 1}$ to align the temporal length of input video for subsequent motion control learning.

## 3.3 Motion Control Learning

With the obtained region-wise trajectory and motion mask, we aim to control motion generation with the input signals. Inspired by the recipe of feature adaptation in controllable image generation (Zhang et al., 2023a), we propose to exploit a lightweight motion encoder to estimate multi-scale features on the input conditions, and utilize these features to adaptively modulate video latent feature in each corresponding scale. To further improve the alignment between input trajectory and generated video, we fine-tune all attention modules in the spatial-temporal transformer blocks of 3D-UNet via using LoRA (Hu et al., 2022).

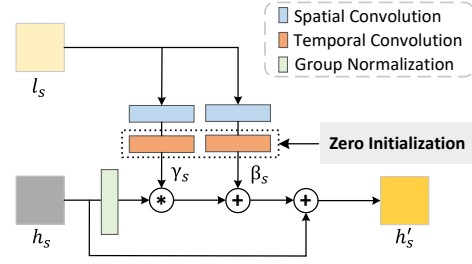

Figure 4: An illustration of adaptive feature modulation in our ReMoCo.

**Adaptive Feature Modulation.** Given the attained region-wise trajectory $\mathbf{T}_s$ and motion mask $\mathbf{M}_{mot}$, we first concatenate them along channel dimension to form the input condition. As shown in Figure 2, a lightweight motion encoder with a series of convolutional layers first encodes the input condition into multi-scale feature maps. In each scale, the learnt feature map is employed to modulate the video latent feature at the same scale in 3D-UNet. Figure 4 depicts an illustration of the adaptive feature modulation by using the feature map $l_s$ in $s$-th scale. Particularly, we estimate the scale $\gamma_s$ and bias $\beta_s$ on $l_s$ via a spatial-temporal convolutional layer. Then, the normalized feature map of the input video latent feature $h_s$ is modulated via $\gamma_s$ and $\beta_s$, and further added back to itself in a skip-connection manner to form the output feature map $h'_s$ as:

$$h'_s = GN(h_s) \cdot \gamma_s + \beta_s + h_s, \tag{8}$$

where $GN(\cdot)$ denotes the group normalization. Note that we implement zero initialization on temporal convolutional layers to initialize $\gamma_s$ and $\beta_s$ as zero at the beginning of training, which guarantees the stability of model optimization.

**LoRA Integration.** To preserve rich motion prior learnt by the pre-trained video diffusion model and elevate the effectiveness of motion control, we employ LoRA layers in all attention modules of spatial-temporal transformer blocks as demonstrated in Figure 2. Specifically, the LoRA parameters $\Delta\mathcal{W}$ act as a residue part of the original weights $\mathcal{W}$ as follows:

$$\mathcal{W}' = \mathcal{W} + \Delta\mathcal{W} = \mathcal{W} + AB^T, \tag{9}$$

where $\mathcal{W}'$ is the fused weights of attention module. $A$ and $B$ are trainable matrices in LoRA layers.

In the training stage, we fix all parameters in the pre-trained 3D-UNet, and only train the lightweight motion encoder and all introduced LoRA layers of the attention modules.

## 3.4 Inference Pipeline of ReMoCo

Our ReMoCo is a user-friendly I2V generation framework for interactive motion control. In the inference stage, as shown in Figure 3, users can readily brush the motion region on the uploaded reference image and draw the trajectory of moving direction as input control signals. In detail, the motion mask can be directly obtained from the user provided brush mask. Given the user trajectory which generally describes the movement of a single pixel, we pad the trajectory value in the $k \times k$ region around the pixel position to match the training paradigm. The padded trajectory in local region is exploited as the input region-wise trajectory. Finally, ReMoCo regulates video denoising with the guidance of the two collaborative control signals through adaptive feature modulation. Both fine-grained and object-level motion control are facilitated by the synergy of the proposed region-wise trajectory and motion mask.

## 4 Experiments

### 4.1 Experimental Settings

**Benchmarks.** We empirically verify the merit of ReMoCo on two benchmarks, i.e., WebVid-10M (Bain et al., 2021) and our proposed ReMoCo-Bench. The **WebVid-10M** dataset consists

| **Input Control** | **DragNUWA** | **DragDiffusion** | **MOFA-Video** | **ReMoCo** |

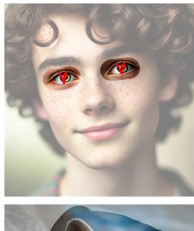

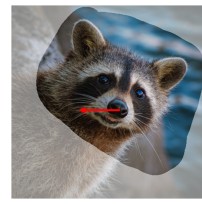

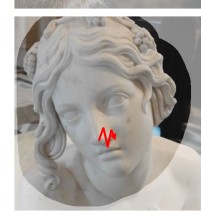

Figure 5: Examples of fine-grained motion control results on ReMoCo-Bench. The input control signals include the reference image, trajectory and motion mask. ***Better viewed with Acrobat Reader for the animated videos.***

of $10.7M$ video-caption pairs. There are $5K$ videos in the validation set and we sample $1K$ videos for evaluation. For each video, trajectories sampled at a ratio of $15\%$ along with the first frame serve as the input condition for fine-grained I2V motion generation. We follow the protocols in recent controllable I2V advance (Niu et al., 2024) and choose the Frechet Video Distance (FVD) (Unterthiner et al., 2019), Frechet Image Distance (FID) (Heusel et al., 2017), and Frame Consistency (Frame Consis.) (Qi et al., 2023) of CLIP (Radford et al., 2021) features as the evaluation metrics on WebVid-10M.

In practical applications, users typically prefer to control video generation through a limited number of representative trajectories, often just one or two. The automatically sampled trajectories employed in WebVid-10M do not adequately represent this scenario, thereby potentially compromising the validity of the evaluation. To address this issue, we introduce **ReMoCo-Bench**, a new benchmark with reference images and user-annotated trajectories, which is tailored for the evaluation of controllable I2V generation. Specifically, we meticulously collect $412$ high-quality reference images from the internet and construct $1.1K$ image-trajectory pairs via human annotation. For each reference image, the annotator is required to brush the motion region and draw the trajectory of movement direction according to the user intention, i.e., fine-grained local part moving or global object moving. As such, the motion control performance can be evaluated from both perspectives. Due to the absence of ground-truth video, FVD and FID metrics are not applicable to ReMoCo-Bench. In addition to Frame Consistency, we utilize the Mean Distance (MD) to measure the alignment between generated motion and input trajectory. Two evaluation protocols are exploited for this target, i.e., MD-Img and MD-Vid. MD-Img is proposed by DragDiffusion (Shi et al., 2024b) which estimates the frame-level mean Euclidean distance between trajectories of input and generated frames. To further validate the video-level trajectory accuracy via MD-Vid, we replace the image correspondence detection model DIFT (Tang et al., 2023) in MD-Img with the video tracking model CoTracker (Karaev et al., 2024), which supplies a more precise trajectory reference.

**Implementation Details.** In ReMoCo, we employ SVD (Blattmann et al., 2023a) as our base architecture. Each training sample is 16-frames video clip and the sampling rate is 8 fps. We fix the resolution of each frame as $320 \times 512$, which is centrally cropped from the resized video. The local region size $k$ is set as 8 and the minimal mask ratio $r_{min}$ is set as $0.95$ determined by cross validation. We set the rank of LoRA parameters as 32. The motion encoder and LoRA layers are trained via AdamW optimizer with the base learning rate $1 \times 10^{-5}$. All experiments are conducted on 6 NVIDIA A800 GPUs with minibatch size $48$.

Table 1: Performances of fine-grained motion control on WebVid-10M.

| Approach | FVD (↓) | FID (↓) | Frame Consis. (↑) |
|----------|---------|---------|-------------------|
| DragNUWA | 96.65 | 13.19 | 0.9888 |
| MOFA-Video | 87.70 | 12.18 | 0.9895 |
| ReMoCo | **59.88** | **10.40** | **0.9895** |

Table 2: Performances of fine-grained motion control on ReMoCo-Bench.

| Approach | MD-Img (↓) | MD-Vid (↓) | Frame Consis. (↑) |
|----------|------------|------------|-------------------|
| DragDiffusion | 14.70 | 13.84 | 0.9947 |
| MOFA-Video | 13.94 | 10.50 | **0.9972** |
| ReMoCo | **10.56** | **8.34** | 0.9962 |

## 4.2 Evaluation on Fine-grained Motion Control

We first evaluate ReMoCo on the fine-grained motion control for I2V generation. The performances on WebVid-10M and ReMoCo-Bench are summarized in Table 1 and Table 2, respectively. Our ReMoCo consistently achieves better performances on WebVid-10M across different metrics. In particular, ReMoCo attains the FVD of 59.88, outperforming the best com-

Table 3: Performances of object-level motion control on ReMoCo-Bench.

| Approach | MD-Img (↓) | MD-Vid (↓) | Frame Consis. (↑) |
|----------|------------|------------|-------------------|
| MOFA-Video | 15.56 | 12.04 | **0.9951** |
| DragAnything | 12.30 | 11.37 | 0.9917 |
| ReMoCo | **10.48** | **8.59** | 0.9943 |

petitor MOFA-Video (Niu et al., 2024) by 27.82. The better FVD indicates the better alignment of data distribution between the generated and ground-truth videos. Such results basically verify the superiority of exploring precise region-wise trajectory to strengthen fine-grained motion dynamic learning. On ReMoCo-Bench, ReMoCo leads to performance boosts against baselines in terms of MD-Img and MD-Vid, showing better alignment between the user input trajectory and synthesized videos. Note that MOFA-Video exploits a two-stage controllable I2V framework that first densifies the input trajectories through conditional motion propagation (CMP), and then calibrates video diffusion process using the estimated dense trajectories. In contrast, ReMoCo learns precise motion patterns by directly referring region-wise trajectory via adaptive feature modulation, thus enhancing the motion-trajectory alignment, as evidenced by the better MD-Img and MD-Vid performances. Besides, the CMP technique in MOFA-Video generally focuses on flow completion in the local region surrounding the input trajectory while neglecting potential movements in other areas. Thus, MOFA-Video tends to synthesize videos with less motion dynamics and obtains slightly higher Frame Consistency (approximately 0.001). To substantiate this, we calculate the average flow magnitude of videos generated by MOFA-Video, which achieves 4.95. In comparison, ReMoCo attains a higher value of 8.95, verifying that our model achieves greater motion variability while maintaining better motion-trajectory alignment.

Figure 5 further showcases three I2V generation results controlled by the user input trajectory and region mask on ReMoCo-Bench. Generally, the videos synthesized by our ReMoCo exhibits more natural movement and better alignment with input trajectory than the baseline methods. For instance, DragNUWA (Yin et al., 2023) suffers from motion misinterpretation issue which wrongly generates videos with camera movement instead of object moving (e.g., the 1st and 2nd cases). The videos generated by MOFA-Video (Niu et al., 2024) usually present unnatural object movement with local part distortion, e.g., the nose of raccoon in the 2nd case. We speculate that such distortion is caused by the lack of global region guidance in MOFA-Video, where the region mask is only employed for flow masking as post-processing. Our ReMoCo, in comparison, integrates the information of motion mask into 3D-UNet on the fly to facilitate the modeling of holistic motion correlation. Thus, the synthesized videos by ReMoCo reflect more rational fine-grained movement.

## 4.3 Evaluation on Object-level Motion Control

Next, we conduct evaluation on object-level motion control for I2V generation. Table 3 lists the performances of different approaches on ReMoCo-Bench. Overall, ReMoCo attains the best performances on the metrics of MD-Img and MD-Vid. Specifically, ReMoCo obtains 10.48 of MD-Img and 8.59 of MD-Vid, reducing the Mean Distance of the best competitor DragAnything (Wu et al., 2024) by 1.82 and 2.78, respectively. The improvements again confirm the merit of leveraging the duet of region-wise trajectory and motion mask for precise motion control. Similar performance trend on Frame Consistency can be also observed in the table.

Figure 6 shows the visual comparison of four object-level motion control results by using different approaches on ReMoCo-Bench. Compared to the baseline methods, videos generated by ReMoCo

**Input Control**        **MOFA-Video**        **DragAnything**        **ReMoCo**

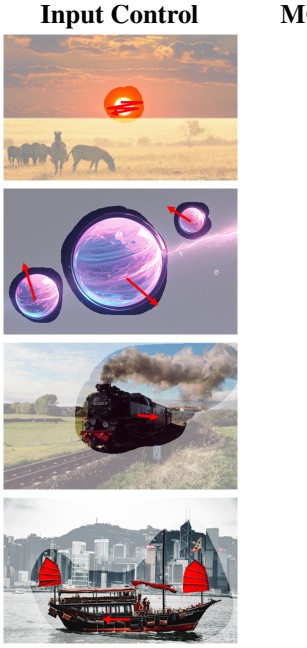

Figure 6: Examples of object-level motion control results on ReMoCo-Bench. The input control signals include reference image, trajectory and motion mask. ReMoCo can successfully handle complicated (e.g., the round trip of sun in the 1st case) and counterintuitive (e.g., the train moving back in the 3rd case) motion-trajectory alignment. ***Better viewed with Acrobat Reader.***

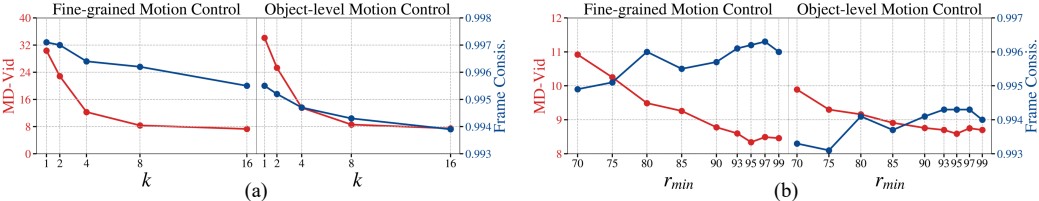

Figure 7: Performance comparisons of MD-Vid and Frame Consistency on ReMoCo-Bench under the settings of both fine-grained and object-level motion control by using different (a) local region size $k$ and (b) minimal mask ratio $r_{min}$ in ReMoCo.

can precisely match the input trajectory and maintain natural object-level motion dynamics. MOFA-Video still faces the challenge of local part distortion (e.g., only the train rear moving back in the 3rd case) and video generation with limited motion dynamics (e.g., the 4th case). Although DragAnything effectively aligns pixel movement with the input trajectory, certain instances (e.g., the 1st and 4th cases) misinterpret the trajectory as camera motion rather than object movement. In contrast, ReMoCo nicely capitalizes on trajectory information to calibrate video denoising and specifies motion category with the region mask, endowing images with high-quality object-level motion.

## 4.4 ABLATION STUDY ON REMOCO

In this section, we perform ablation study to delve into the design of ReMoCo for controllable I2V generation. Here, all experiments are conducted on ReMoCo-Bench for performance comparison.

**Local Region Size.** We first investigate the choice of local region size $k$ for region-wise trajectory design in our ReMoCo. Figure 7(a) compares the performances of MD-Vid and Frame Consistency on both fine-grained and object-level motion control by using different $k$. The variation of Frame Consistency is minor (less than $0.01$) across different settings, and the MD-Vid decreases when using larger $k$. When $k$ is small (e.g., $1$ or $2$), the kept trajectories are less in each local region and the control signals are weaken for motion control, leading to the inferior trajectory matching performance. Meanwhile, the improvement of MD-Vid is marginal when increasing $k$ to 16. Specifically, using large $k$ will extend the input trajectory over a large region, which affects the fine-grained

Control      k=16      **k=8**      Control      k=1      k=2      k=4      **k=8**

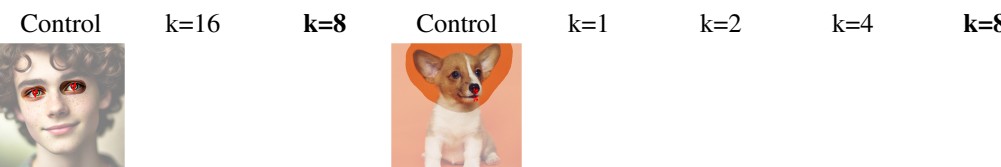

Figure 8: Visualization of controllable I2V generation results with different local region size $k$ in ReMoCo. ***Better viewed with Acrobat Reader for the animated videos.***

motion control. Accordingly, we exploit $k = 8$ to extract the region-wise trajectory as the motion condition. Figure 8 further illustrates the I2V generation results with different $k$. As shown in this figure, the synthesized videos with $k = 8$ present more natural motion dynamics and more precise motion-trajectory alignment. Moreover, the unnatural fine-grained motion as shown in the case when $k = 16$ validates our analysis on the influence of overlarge region size.

**Minimal Mask Ratio.** To explore the effect of minimal mask ratio $r_{min}$ in trajectory selection stage, we then measure the motion control performance by conducting different $r_{min}$ in Figure 7(b). Overall, Frame Consistency is not sensitive when changing $r_{min}$ on both fine-grained and object-level motion control settings. Meanwhile, the performance of MD-Vid becomes better with the increase of the mask ratio at the beginning. The results are expected since using small $r_{min}$ will sample more trajectories for model training, which enlarges the gap between training and real-world inference (i.e., only using one or two trajectories). Conversely, employing a large value of mask ratio (e.g., $0.99$) could make it difficult to optimize networks with scarce trajectory signals. Therefore, we empirically set $r_{min}$ as $0.95$ to obtain the best motion-trajectory alignment in the generated videos.

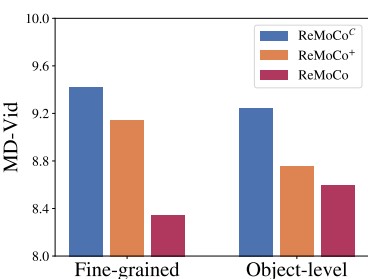

Figure 9: MD-Vid ($\downarrow$) among different multi-scale feature injection approaches on ReMoCo-Bench.

**Multi-scale Feature Injection.** We also investigate different multi-scale feature injection strategies in ReMoCo. Figure 9 details the MD-Vid performance comparisons among different variants of our ReMoCo. **ReMoCo$^C$** concatenates the multi-scale features learnt by motion encoder with the video latent features along channel dimension in each scale. **ReMoCo$^+$** replaces the channel-wise feature concatenation in ReMoCo$^C$ with the feature summation. In comparison, our proposal (**ReMoCo**) injects the control signals into 3D-UNet via the adaptive feature modulation. Overall, ReMoCo exhibits better MD-Vid performances against other two variants. In direct feature aggregation methods such as concatenation or summation, information exchange requires strict spatial-temporal alignment between each other. In contrast, there is no such requirement for feature modulation, as it indirectly utilizes estimated scale and bias for feature regulation. Consequently, such feature injection approach demonstrates enhanced capacity to extract relevant information from input signals, potentially leading to improved motion control performance.

## 5   Conclusions

This paper explores the motion condition formulation and the motion-trajectory alignment in diffusion models for controllable I2V generation. In particular, we study the problem from the viewpoint of integrating accurate motion control signals into video denoising to regulate motion generation. To materialize our idea, we have devised ReMoCo, which leverages the region-wise trajectory and motion mask as the condition to calibrate video generation in a feature modulation manner. The region-wise trajectory preserves the original optical flow information in each local region, characterizing the fine-grained motion details. The motion mask derived from the optical flow maps presents holistic motion and aims to identify exact target motion category. The collaboration of two signals regulates video denoising for natural motion synthesis with precise motion-trajectory alignment. Moreover, we have carefully construct a new benchmark, i.e., ReMoCo-Bench, with $1.1K$ real-world user-annotated image-trajectory pairs for the evaluation of both fine-grained and object-level motion control. Extensive experiments on WebVid-10M and ReMoCo-Bench validate the superiority of our proposal over state-of-the-art approaches.

## 6 ETHICS STATEMENT

The primary of this paper is to introduce a controllable image-to-video diffusion model for general individuals to animate particular object motion. It is important to note that the visual contents of our generated video are aligned with those of input reference image. Even though there could be some ethical concerns for input image, employing an additional content safety checker to filter reference image can potentially resolve this issue. We uphold the highest ethical standards in the construction of our ReMoCo-Bench, and believe that all the contents in the dataset are appropriate while respecting relevant privacy rights in data collection procedure.

## 7 REPRODUCIBILITY STATEMENT

We have introduced the model construction and implementation details in the paper. To enhance the reproducibility of our approach, we attach the core code of ReMoCo in the supplementary material with detailed explanations.

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

Table 4: Human evaluation of user preference ratios (%) over both fine-grained and object-level motion control across different approaches on ReMoCo-Bench.

| Evaluation Items | Fine-grained Motion Control | | | Object-level Motion Control | | |
|---|---|---|---|---|---|---|
| | DragDiffusion | MOFA-Video | ReMoCo | MOFA-Video | DragAnything | ReMoCo |
| Motion Quality (↑) | 3.12 | 21.88 | **75.00** | 12.50 | 18.75 | **68.75** |
| Temporal Coherence (↑) | 6.25 | 40.63 | **53.12** | 25.00 | 15.63 | **59.37** |
| Trajectory Alignment (↑) | 9.37 | 18.75 | **71.88** | 15.62 | 21.88 | **62.50** |

## A    APPENDIX: MORE DETAILS OF REMOCO-BENCH

The proposed ReMoCo-Bench consists of 412 high-quality reference images and corresponding $1.1K$ user-annotated trajectories. We collect the reference images with different visual contents, including animal, human, vehicle, etc. There are 72 images sampled from the public DragBench (Shi et al., 2024b) and we further extend it with 340 additional images. Specifically, all the self-collected images about human are automatically generated by DALL·E3 (Betker et al., 2023) to avoid the potential legal concerns. The remaining self-collected images are real photos which are first crawled on the Pexels platform and then filtered according to the visual quality. For each reference image, the annotator is required to brush the motion region and draw the movement trajectory according to user intention (i.e., fine-grained local part moving or global object moving). During trajectory annotation, all annotators are encouraged to ensure the trajectory diversity, including some complicated trajectories. Finally, the benchmark is annotated with 460 image-trajectory pairs for fine-grained motion control evaluation, and 680 image-trajectory pairs for object-level motion control evaluation, respectively. Figure 10 and Figure 11 further illustrate several visual examples (reference image, trajectory and motion mask) from ReMoCo-Bench for the two evaluations.

## B    APPENDIX: HUMAN EVALUATION

In addition to the evaluation over automatic metrics, we also conduct human evaluation to investigate user preferences from three perspectives (i.e., motion quality, temporal coherence and trajectory alignment) across different controllable I2V approaches. In particular, we randomly sample 200 generated videos from both fine-grained and object-level motion control for evaluation. Through the Amazon MTurk platform, we invite 32 evaluators, and ask each evaluator to choose the best one from the generated videos by all models given the same inputs.

Table 4 shows the user preference ratios across different models on ReMoCo-Bench. Overall, our ReMoCo clearly outperforms all baselines in terms of the three criteria on both fine-grained and object-level motion control. The results demonstrate the advantage of leveraging complementary region-wise trajectory and motion mask to benefit video synthesis with natural motion, desirable temporal coherence and precise motion-trajectory alignment.

## C    APPENDIX: OFFLINE PROJECT PAGE

We build an offline project page for our ReMoCo in the "ReMoCo.github.io" folder, and package it into supplementary material. Please click the file of "index.html" in the folder with the Chrome or Firefox browser for more vivid video presentation.

## D    APPENDIX: CODE RELEASE

Moreover, we package the core code of our ReMoCo in the "ReMoCo-Code" folder of supplementary material. Please refer to the example source code and README in the folder for more details.

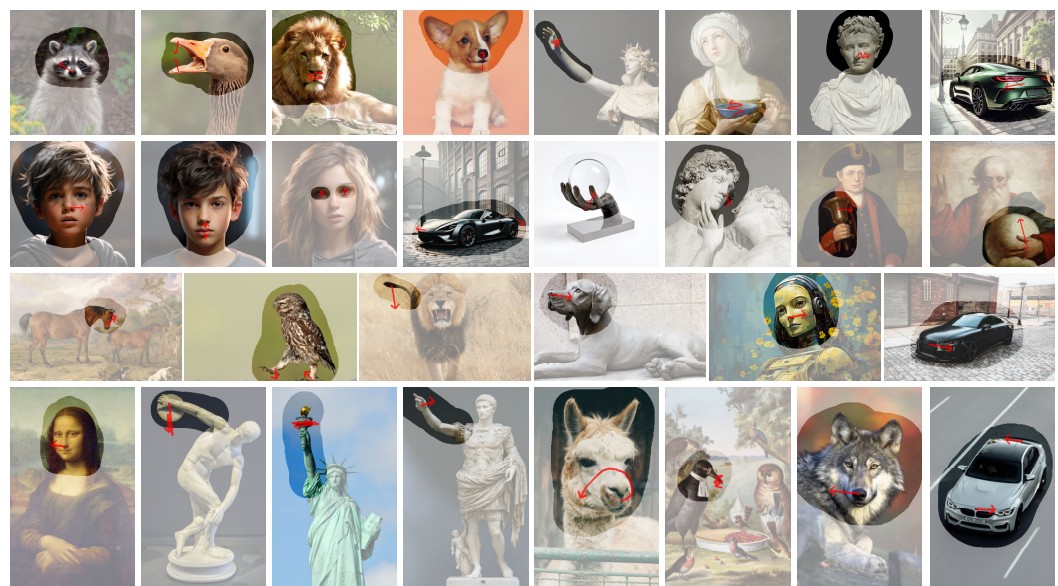

Figure 10: Visual examples from ReMoCo-Bench for fine-grained motion control evaluation. Each reference image is annotated with trajectory and motion mask.

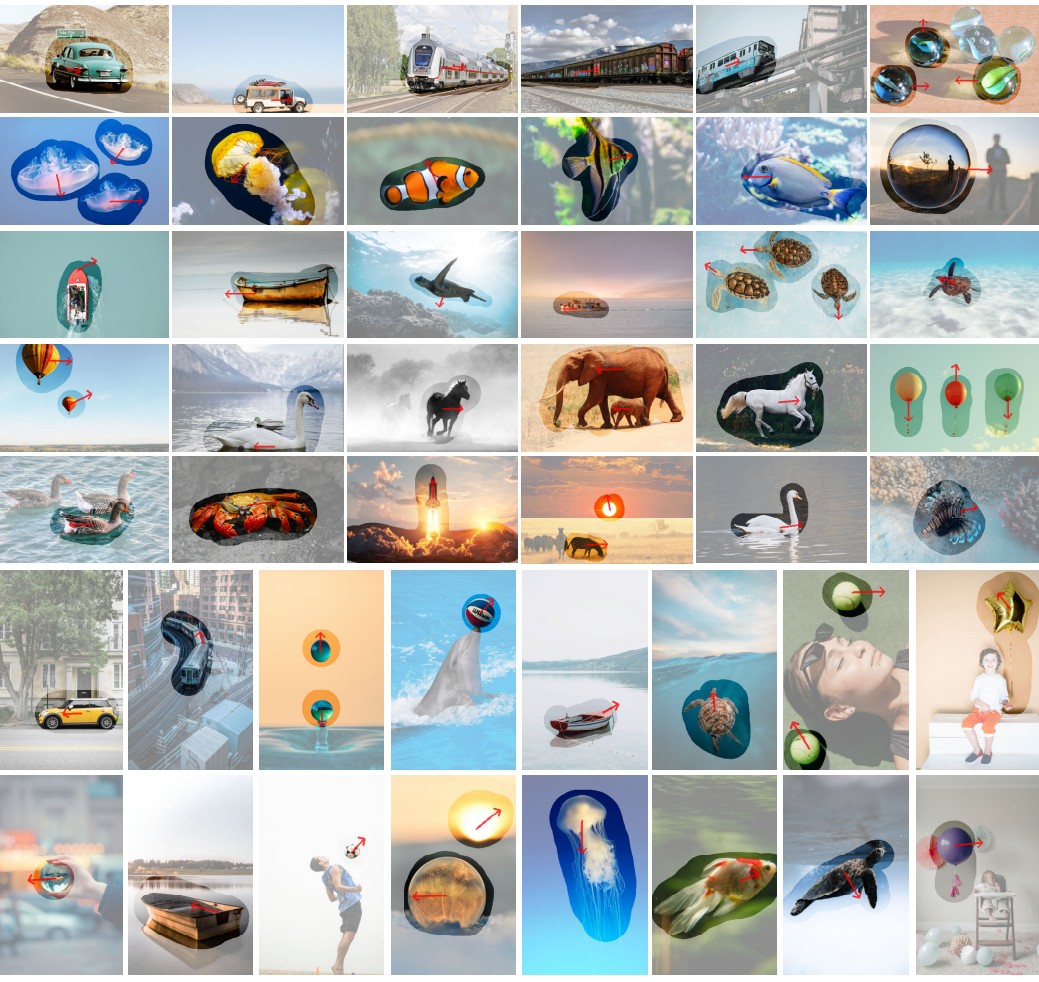

Figure 11: Visual examples from ReMoCo-Bench for object-level motion control evaluation. Each reference image is annotated with trajectory and motion mask.

