# OpenReview forum: "Region-wise Motion Controller for Image-to-Video Generation"
_ICLR.cc/2025/Conference — ICLR 2025 Conference Withdrawn Submission_

### Official Review · Reviewer_CpUo · 2024-11-01

**Soundness:** 3
**Presentation:** 2
**Contribution:** 3
**Rating:** 6
**Confidence:** 4

**Summary:**

This paper presents a framework for animating images with interactive motion control, addressing the limitations of fine-grained movement control in image-to-video (I2V) generation, especially where object and camera motions are intertwined. To overcome these challenges, they introduce ReMoCo, a region-wise motion controller that utilizes precise region-based trajectories and motion masks to facilitate fine-grained motion synthesis and distinguish between object and camera motion.

**Strengths:**

1.	Other methods sample sparse motion trajectories from optical flow maps and process them by Gaussian filter. However, the flow approximation introduced by Gaussian filtering inevitably results in inaccuracies in fine-grained motion details, limiting the model's ability for precise motion control. This paper proposes a Region-wise Trajectory approach, which spatially samples multiple local regions in the video’s optical flow maps and directly uses the trajectories in these sparse regions as input trajectory conditions. In this way, no flow approximation is included in the region-wise trajectory, allowing it to adequately capture local fine-grained motion details.

2.	Additionally, a masking strategy is proposed to control motion regions, thereby solving the problem of existing methods being unable to distinguish between camera motion and object motion.

3.	Adaptive feature modulation is used to inject multi-scale control information, eliminating the need for spatiotemporal alignment between control features and video features.

4.	The performance shows a significant improvement compared to the baseline.

**Weaknesses:**

Since it is mentioned that the problem of other methods being unable to distinguish between camera motion and object motion has been solved, the presented cases demonstrate good control over object motion. However, there are no showcased cases with camera motion. It would be helpful to demonstrate the control capability over camera motion as well.

**Questions:**

None

---

### Official Review · Reviewer_XKrZ · 2024-11-02

**Soundness:** 2
**Presentation:** 3
**Contribution:** 2
**Rating:** 3
**Confidence:** 4

**Summary:**

The paper presents an approach to motion-controlled image-to-video generation, where motion is controlled by user-specified trajectories.
To encourage object motions rather than mere camera motions, users can additionally specify motion regions to limit the area of object movement.
With this problem formulation, the authors developed a dataset creation pipeline that automatically extracts sparse trajectories (each trajectory is represented by an 8x8 patch for each frame) and motion regions (by thresholding average per-pixel optical flow across frames) from the videos extracted from WebVid-10M.
After that, a pre-trained SVD inflated with a motion encoder is trained. Specifically, the feature outputs from the motion encoder are injected into SVD through adaptive normalization, which demonstrated better performance than control-net-like injections such as feature summations. Additionally, attention layers are finetuned with LoRA.
Evaluation on WebVid-10M and their own created dataset (ReMoCo-Bench) shows improvements in object-level and fine-grained motion controls over existing baselines.

**Strengths:**

- The authors proposed a new task formulation that can intuitively control the motions of videos by specifying trajectories and moveable regions. Although this problem formulation has already been proposed by DragDiffusion, it is the first work in the context of video diffusions.

- The authors demonstrate that adaptive normalization performs better than control-net-like conditioning like feature summations through quantitative experiments. Although adaptive normalization has been already implemented by previous works (e.g., DragNUWA), actually showing its efficacy over feature summations for the first time is essential. (However, note that concurrent works such as Tora also provide similar experiments.)

- The authors built an evaluation dataset that contains 1.1 K human-annotated datasets. The authors haven't mentioned anything, but if authors can release the dataset publicly, that would be one of the contributions given there are no standard evaluation datasets in this community.

- The authors provide a supplementary website with a sufficient number of examples with various trajectories, which is helpful for reviewing the paper.

**Weaknesses:**

Although I appreciate the author's engineering efforts to build the user-friendly model, I have some concerns regarding their technical motivation, datasets, evaluation, and novelty which led me to vote for rejection.

Motivation:

One of the authors' key claims is that applying Gaussian filtering to input trajectory points (as done in previous works) prevents achieving fine-grained motion controllability. However, this doesn't seem to be accurate:
- I am not convinced that Gaussian filtering prevents fine-grained motion controllability. Gaussian densities can tell the center coordinates of trajectory points, which intuitively may not ruin fine-grained motion controllability. I would appreciate it if the authors could demonstrate why Gaussian filtering should be removed through rigorous experiments. (please see below for suggestions of experiments).
- During inference, the authors embed input trajectory points by duplicating them into 8x8 patches per frame, which is essentially the same as applying Gaussian filtering over 8x8 patches and I don't see any significant advantage of removing Gaussian filtering in the paper.
- Could the authors demonstrate the loss of fine-grained controllability by replacing 8x8 patches with Gaussian filtered points (where each point is represented by 8x8 Gaussian density), both during training and inference? This formulation comes from DragNUWA. Without a clear drop in performance, I think this reasoning does not seem convincing.

Datasets:

- If I understand correctly, motion regions placed on the first frame specify the moveable regions "across" frames. However, most of the examples shown in Figure 10 and Figure 11 tightly enclose objects and do not account for regions that are affected at later frames (e.g., regions where the target objects eventually arrive). Isn't there a domain gap between the authors' task formulation and the datasets?

Evaluation:

- The authors compare their approach with other trajectory-conditioned image-to-video baselines. However, the experiments do not seem to be a fair comparison, where the author's model is given extra input conditioning: "movable region", while other baselines are only given trajectories that do not tell movable region. To keep a fair comparison, I would suggest placing zero motion trajectories outside the moveable regions in baseline approaches.  One of the authors' key claims is that previous baselines may introduce unwanted camera motions, however, it can be resolved by placing extra zero-vectors on immovable regions like how the movable regions are introduced in the authors' work.

- The authors evaluate their approaches on WebVid-10M with FVD/FID/Frame Consists. However, these metrics are for measuring visual quality and should not represent motion fidelity. Could the authors provide MD-Vid as done in the other dataset to measure motion fidelity?

- Could authors add DragNUWA to the evaluation on ReMoCo-Bench as well? DragNUWA is a key baseline that does not distinguish object-level and fine-grained level control like the authors' works. Hence, comparing with DragNUWA is important and should not be missed.

- The ablation study on local region size does not seem convincing. The authors claim that k=16 has worse fine-grained motion controllability than k=8, but the claim seems conflicting with Figure 7 where k=16 quantitatively achieves better motion fidelity. I appreciate it if the authors could provide convincing quantitative results.

Training dataset creation:
- If I understand correctly, training datasets are created by masking out the "same" pixel locations across frames regardless of object motions. This is different from previous works (e.g., DragNUWA, DragAnything, ..) that track moving points across frames to create input trajectory conditioning. Could the authors provide reasons for not taking the conventional approach?
- The above formulation may hurt trajectories with large motion as it can not track points over an 8x8 patch. However, the results of large motions are not demonstrated in the paper. Could the authors consider adding results with larger motions?

Novelty:

- Given the above discussion regarding Gaussian filtering and a lack of fair evaluation, the technical novelty is not clear. Especially, I am wondering exactly what contributes to fine-grained motion control. I would appreciate it if the authors could clarify that by comparing with baselines or referring to ablation studies.

**Questions:**

I would appreciate it if the authors could address my concerns in weakness.

---

### Official Review · Reviewer_kHbL · 2024-11-03

**Soundness:** 3
**Presentation:** 3
**Contribution:** 2
**Rating:** 5
**Confidence:** 4

**Summary:**

This paper presents a novel approach (ReMoCo) for interactive motion control in I2V generation. Unlike traditional methods that use Gaussian-filtered point-wise trajectories, ReMoCo employs region-wise trajectories and motion masks, allowing for precise control over fine-grained movements and clear differentiation between object and camera motion. ReMoCo first estimates flow maps from training videos, then samples region-based trajectories, preserving local flow details to capture intricate motion. Additionally, a motion mask highlights overall motion dynamics, enhancing controllable video denoising.

**Strengths:**

Well written and easy to understand. The paper proposes a region-wise motion control based on motion direction and masks, which aids in achieving fine-grained control in video generation with friendly user interaction.

**Weaknesses:**

The paper emphasizes its success in addressing the issue of incomplete decoupling of object and camera motion present in other methods. However, the experiments and demos provided in the paper provided involve static backgrounds without camera movement, suggesting that this method may only be suitable for fixed-view object motion. This setup makes it difficult to demonstrate a clear advantage in decoupling over other approaches.

**Questions:**

1. There are some typos, such as the highlighted annotation in Table 1.
2. See weakness. I would like to know if the method presented in this paper can be applied to video generation driven by camera trajectories in conjunction with the motion control proposed in the paper.
3. I really like the demos provided on the supplementary material webpage, as it showcases the contributions of this paper. However, I noticed some flaws and instabilities in these videos. For instance, in “Interactive Motion Control for Image-to-Video Generation,” the surface of the moving planet exhibits persistent noise fluctuations. In “Gallery: Object-level Motion Control,” the bus in the third row sticks to the white balls behind it when moving forward, but not when reversing. Additionally, in the last row, the stem of the sunflower is affected by the background. I understand that it can be extremely difficult to completely resolve these issues with existing methods, but I am curious if you have any measures in place to mitigate such flickering and background sticking situations.

---

### Official Review · Reviewer_ZSxz · 2024-11-04

**Soundness:** 3
**Presentation:** 3
**Contribution:** 3
**Rating:** 6
**Confidence:** 3

**Summary:**

To enhance the controllability of fine-grained movement in current image-to-video frameworks, this paper introduces the Region-wise Motion Controller (ReMoCo), which utilizes precise region-wise trajectories and motion masks to regulate fine-grained motion synthesis and identify specific target motions. Specifically, during training, flow maps are estimated from each training video, and region-wise trajectories are sampled within local areas along with corresponding motion masks. These additional conditions (region-wise trajectory and motion mask) are integrated in a feature modulation manner to refine motion synthesis.
The authors also introduce a new benchmark, ReMoCo-Bench, designed to evaluate both fine-grained and object-level motion synthesis in image-to-video (I2V) generation. Extensive experiments are conducted on the WebVid-10M dataset and ReMoCo-Bench, demonstrating the effectiveness of the proposed approach.

**Strengths:**

1. The paper is clearly written and easy to follow.

2. The proposed design of the region-wise motion controller, along with the training strategy used to build the training data, appears well-thought-out. The provided examples demonstrate high-quality results compared to baseline methods.

3. The authors have made the code implementation and an offline project page available, offering additional examples and enhancing the reproducibility of the work.

**Weaknesses:**

1. There are some relevant baselines, such as Motion-I2V [1] (in the motion drag section) and Drag-a-Video [2], which support local region motion guidance. Including these methods in the experiments for comparison would strengthen the evaluation of the proposed approach.

2. It is unclear why the baseline methods differ across Tables 1, 2, and 3. Would it be possible to evaluate all baseline methods consistently across all three tables to provide a more comprehensive comparison?

3. Since the method operates on a masked region, it appears to support only local region motion, with the background remaining static. This setup effectively limits the approach to a "static camera" scenario, restricting its applicability in more general settings. Although the abstract claims that the method can "identify the exact target motion category (i.e., object or camera moving)," the reviewer did not find any examples of camera motion in either the paper or the project page. It would be valuable if the authors could discuss potential approaches to overcome this limitation.

4. Since there is no camera motion involved in this approach, it may be beneficial to include DragGAN as an additional baseline. Although DragGAN [3] is primarily an image editing framework and may exhibit lower pixel-level consistency during the progress, it has demonstrated strong performance and control accuracy, making it a competitive alternative worth comparing.

5. It would be helpful to include a discussion in the paper on the limitations of the approach and possible directions for future work.

[1] Motion-I2V: Consistent and Controllable Image-to-Video Generation with Explicit Motion Modeling

[2] Drag-A-Video: Non-rigid Video Editing with Point-based Interaction

[3] Drag Your GAN: Interactive Point-based Manipulation on the Generative Image Manifold

**Questions:**

Please refer to the weakness section.

---

### Note · Authors · 2024-11-14

I have read and agree with the venue's withdrawal policy on behalf of myself and my co-authors.